# Assessing the knowledge, attitudes, practices, and perspectives of stakeholders of the deworming program in rural Rwanda

Fernand Rwamwejo[1]*, Grace Iliza Ndatinya[2], Madalitso Ireen Mkata[3], Amy Blauman[4], Denis Regnier[5], Sarah Cruz Lackner[6]

1 Director General's office, Rwanda Biomedical Center, Kigali, Rwanda, 2 Department of Community Health and Social Medicine, University of Global Equity, Kigali, Rwanda, 3 Department of Policy and Partnerships, Partners in Health, Freetown, Sierra Leone, 4 Programme unit, World Food Programme, Kigali, Rwanda, 5 Department of Humanities and Social Sciences, University of Global Health Equity, Kigali, Rwanda, 6 M&E unit, World Food Programme, Kigali, Rwanda

* rwamwejo@gmail.com

**Data Availability Statement:** All relevant data are within the manuscript and its Supporting Information files.

## Abstract

### Background

Worm infections are among the most prevalent neglected tropical diseases worldwide. Schistosomiasis and soil-transmitted helminths infections, most common worm infections affecting Rwandan school-aged children, are addressed by the national deworming program since 2014. To date, no published studies have assessed the knowledge, attitudes, and practices of the key implementers of the national deworming program conducted at village and school level in Rwanda. This study aimed to assess key stakeholders' knowledge, attitudes, practices, and perspectives about the decentralized national deworming program.

### Methods/Principal findings

We carried out a quantitative, cross-sectional study with complementary in-depth interviews in two districts of Rwanda in June 2021. From the 852 surveyed community health workers and teachers, 54.1% had a knowledge score considered good ($\geq$80%). The mean knowledge score was 78.04%. From the multivariate analysis, lack of training was shown to increase the odds of having poor knowledge (OR 0.487, 95% CI: 0.328–0.722, p <0.001). The in-depths interviews revealed poor water access and hesitance from caregivers as perceived challenges to the success of the deworming program.

### Conclusion

Our findings demonstrate the importance of training community health workers and school-teachers on worm infections as they are the key implementers of the deworming program. This would enhance their capacity to provide health education and sensitization on misconceptions and misinformation towards deworming. Moreover, research is needed to assess the impact of poor access to water, sanitation and hygiene facilities on the prevalence of worm infections in Rwanda.

**Funding:** The author(s) received no specific funding for this work.

**Competing interests:** No authors have competing interests

## Author summary

Worm infections affect over 2 billion people worldwide, and the most common are schistosomiasis and soil-transmitted helminth infections. In Rwanda, the prevalence of soil-transmitted helminth infections can be as high as 45%, while the prevalence of schistosomiasis is 2.7% among school-aged children.

The government of Rwanda continues to develop initiatives to eliminate worm infection, among which, the national decentralized deworming program, implemented at village and school level.

In this program, community health workers collaborate with teachers and local leaders to administer deworming tablets, community sensitization and health education. To date, no published studies have been conducted to understand the experiences and perspectives of key stakeholders in the national deworming program in Rwanda. We conducted a quantitative analysis of the Knowledge, Attitudes and Practices (KAP) of teachers and community health workers, and a qualitative assessment of the perspectives of local leaders in two of the most food insecure districts of Rwanda. The results show a gap in the knowledge of community health workers and teachers on worm infections. The study also revealed that there were some misconceptions about deworming tablets, which highlights the need to train key implementers of the program for them to transfer their knowledge to the broader community.

## Introduction

Worm infections affect more than 2 billion people worldwide, with soil-transmitted helminth infections (STH) and schistosomiasis being the two most common types [1]. In 2022, about 1.5 billion people were affected by STH, and 200 million by Schistosomiasis [2,3]. The infections disproportionally affect people living in impoverished situations in low and middle income countries (LMIC) [4], with over 90% of Schistosomiasis cases being recorded in sub-Saharan Africa [2,3].

STH and Schistosomiasis are transmitted through contaminated water or soil, either orally or through punctured skin [4–6]. People living in impoverished situations are at higher risk due to lack of adequate access to clean water, basic sanitation and hygiene facilities, and footwear [4,7]. People living in rural communities where agriculture and fishing are commonly practiced are at high risk. The infections can affect people from all age groups, but children were found to be more affected by numerous long-term effects. Worm infections cause numerous intestinal symptoms and chronic illness, which frequently lead to decreased growth, and cognitive impairment resulting in long term diminished productivity [8–11]

In Rwanda, Schistosomiasis and STH are the most common neglected tropical diseases [12]. Occupational exposure through agriculture and water bodies is a common risk factor as 74% of households (primarily rural) are engaged in agriculture, water bodies such as lakes are subject to urban waste, household waste and insufficient sewage water treatment. [13]. A nationwide study conducted in 2008 showed that 65.8% of school-aged children had STH, with a higher prevalence in rural districts compared to Kigali [14].

A follow-up study conducted in 2014 showed a slight decrease in the prevalence, with an overall 45% of school-aged children infected with STH [15]. The 2008 mapping revealed a 2.7% nationwide prevalence of schistosomiasis [12] with a district level prevalence ranging from 0 to 69.5% among school children.

Mass Deworming Administration (MDA) is the recommended intervention to reduce and manage STH and schistosomiasis in conjunction with programs increasing access to clean water, hygiene, and sanitation facilities [4]. In 2014, the Rwandan Ministry of Health, supported by WHO and the World Food Programme (WFP), adopted a nation-wide deworming program to address worm infections. WFP has been implementing the Home-Grown School Feeding program (HGSF) since 2016 in four of Rwanda's most food insecure districts: Nyaruguru and Nyamagabe in the southern province, as well as Karongi and Rutsiro in the western province. This program includes a comprehensive package of complementary activities aiming to improve quality of education, academic performance, nutrition, and health. WFP directly supports deworming activities in schools as part of the biannual integrated national maternal and child health week campaign, led by the Rwanda Biomedical Center [16]. The national deworming programs, targeting children and communities, are conducted by community health workers (CHWs) and schoolteachers through outreach campaigns, with the local community leaders supervising the implementation. Understanding the knowledge, attitudes, and practices (KAP) of teachers and CHWs and the perspectives of local leaders is necessary to ensure adequate training, support, and supervision of these key implementers [17]. To our knowledge, there is a dearth of studies assessing the KAP of teachers and CHWs, as well as the perceptions of local leaders about deworming in Rwanda. Our study addressed this gap and provided insights into the Rwandan national deworming program.

## Methods

### Study setting

The investigation took place in Nyamagabe and Rutsiro districts, two of the four districts where the WFP's Home-Grown School Feeding (HGSF) program is implemented. In 2021, 330,000 people resided in Nyamagabe, including 1,608 CHWs and 103 primary schools, 25 of which were implementing the HGSF program [18,19]. Rutsiro had a total population of 324,654 people including 1,449 CHWs; and 87 primary schools, 21 of which were implementing the HGSF program [20,21].

### Study design

The study utilized both quantitative and qualitative study methods. A cross-sectional quantitative survey was used to assess the KAP of schoolteachers and CHWs about deworming, and in-depth interviews (IDIs) were used to collect the perspectives of local leaders about the deworming program implemented at village level, using a phenomenological design.

### Sample

**Survey.** Teachers from schools and CHWs working in villages where the HGSF program were implemented in the study sites were included in the study. Those who did not participate in the deworming program were excluded.

**In-depth interview.** Local leaders in the study sites who participated in the national deworming program were the target sample. These are village leaders, who are elected to serve and represent residents in solving local issues, ensure security as well as the implementation of government programs.

### Sample size and sampling

**Survey.** Cluster sampling was used. The sample size was calculated using karimollah formula [22] and multiplied by the design effect to adjust for the design effect of the sample design. A

sample size of 326 teachers was needed to achieve a 95% confidence level, 0.05 margin of error. There were 755 teachers in 46 schools where the HGSF program is implemented. Given that 52.3% (395) of teachers were in Nyamagabe and 47.7% (360) in Rutsiro, we randomly selected 13 schools from Nyamagabe district and 12 schools from Rutsiro district. All teachers who fulfilled the selection criteria in the selected schools were invited to participate in the survey.

$$n = \left( t^2 \times \left[ \frac{p \times q}{d^2} \right] \right) \times DEFF$$

$$n = \frac{n_0 N}{n_0 + (n-1)}$$

$$Final\ N = \frac{Number\ of\ participants\ needed}{1 - NRR}$$

**N**: Total sample in 2 districts
**t**: linked to 95% confidence interval for cluster sampling = 1.96
**p**: expected prevalence (fraction 1) = 0.5
**q**:1-p (expected non-prevalence) = 0.5
**d**: relative desired precision = 0.5
**DEFF**: Design Effect = 1.5
**n**: sample size
**NRR**: Non-response rate = 5%

Using the same formula, a sample size of 484 CHWs was needed. There were 3057 CHWs in both districts. Given that 52.6% (1608) of CHWs were in Nyamagabe and 47.4% (1449) in Rutsiro, 15 cells (administrative unit above village level and below sector level in Rwanda) from Nyamagabe district and 10 cells from Rutsiro district were randomly selected. All CHWs who fulfilled the selection criteria in the 25 selected cells were invited to participate in the survey.

**In-depth interview.**   Sample size was determined by theoretical saturation, which in qualitative research is described as the point where no new information is arising from the data [23]. Purposive sampling was used. This is a non-probability sampling method, where elements are chosen based on the judgment of the researcher [24].

## Data collection procedures

**Survey.**   Verbal approvals were obtained from school directors. Data collectors went to individual schools to administer the survey to the teachers.

CHWs were contacted through phone calls according to the contact information provided by health district coordinators. Data collectors met with the CHWs at their preferred locations according to the appointment to administer the survey.

Informed consent was obtained after detailed explanation of the study was provided. The data collectors read the questions to individual participants in Kinyarwanda (the local language) and recorded the responses directly on Open Data Kit (ODK) through tablets. The survey took approximately 20 minutes to complete.

**In-depth interview.**   Local community leaders were contacted through the district health coordinators. They were first contacted through phone calls and data collectors met with them at the agreed appointment time and locations.

Informed consent and permission to record were obtained after detailed explanation of the study was provided. The interviews were conducted in a private office in Kinyarwanda; only the researcher and participant were in the room during the interview.

### Data collection tools

**Survey.** A questionnaire (S1 Appendix) was developed based on previous similar KAP studies [25].

The questionnaire had four parts. Part 1 collected some basic demographic information. Part 2 contained 16 "yes", "no" or "I don't know" knowledge questions, with nine related to schistosomiasis and seven related to STH. Part 3 contained ten 4-point Likert scale statements, with five related to the respondents' attitude on Schistosomiasis and five on STH [26]. The options for these statements were: "strongly agree", "agree", "disagree" and "strongly disagree". Part 4 contained six 4-point Likert scale statements related to their practices. The options for these statements were: "never", "rarely", "often" and "always".

**In-depth interview.** A semi-structured interview guide (S2 Appendix) was developed. Six main open-ended questions with probes were included to capture the experiences, challenges, best practices and recommendations of local leaders in the implementation of the national deworming program.

Both the survey and semi-structure interview guide were developed in English then translated to Kinyarwanda and back translated to English. Modifications were made based on the feedback from pre-testing before the final version was administered. Pre-testing of the tool was done on a population with similar characteristics as the respondents.

### Measures

Four key measures were included in this study:

1. Knowledge level, subdivided into STH and Schistosomiasis

2. Attitude toward MDA program, subdivided into STH and Schistosomiasis (Attitude Category): Each attitude statement was measured on a four-point Likert scale for those who "strongly agree, agree, disagree, and strongly disagree". For each statement of attitudes, "strongly agree" and "agree" were categorized into one group, while "disagree" and "strongly disagree" were categorized into another group. The percentages of "agree/strongly agree" and "disagree/strongly disagree" were used for analysis.

3. Practices on worm infection prevention (Practice Category): The practice category was also measured on a four-point Likert scale for "never, rarely, often, and always". For practice questions, "often" and "always" were grouped into one category and "rarely" and "never were grouped into another category. The percentages of "often/always" and "never/rarely" were used for analysis.

4. Perspectives about the deworming program

### Data management and analysis

Descriptive analyses were conducted for demographic information and the KAP results. The knowledge score was calculated as the percentage of correct answers divided by questions answered. The overall knowledge was categorized as "good" if the score was between 80 and 100%, and poor if less than 80% [27]. The attitude statements were presented as the percentage of choosing "agree" and "strongly agree" versus the percentage of choosing "disagree" and "strongly disagree". The practices statements were presented as the percentage of choosing "never" and "rarely" versus the percentage of choosing "often" and "always".

Fisher's exact test was used to analyze the association between knowledge level and demographics, between knowledge level and attitude category, and between knowledge level and practice category [28].

All demographic variables with P<0.10 in the bivariate analysis were then included in the logistic regression model for analyzing factors contributing to knowledge level, attitude and practices.

All the statistical tests were performed using SPSS v.27.0.1. All results of the statistical tests with a p-value <0.05 were deemed to be significant.

The IDIs were transcribed and translated. The codebook was developed based on the interview responses inductively and iteratively by three investigators. All transcripts were coded based on the final codebook first individually then together as a team. Thematic analysis was used to identify emerging themes. Qualitative analyses were conducted using Dedoose software [29].

### Ethical consideration

This study was approved by the University of Global Health Equity institutional review board.

## Results

### Survey results

**Sample demographic characteristics.** A total of 852 people were surveyed, among which 509 (59.7%) were CHWs and 343 (40.3%) were teachers. Out of the 852 respondents, 542 (63.6%) were female, 382 (44.8%) had received training on deworming, 448 (52.6%) were from Nyamagabe and 404 (47.4%) from Rutsiro. Among the 382 participants who had received training, there was a significant difference between CHWs and teachers who had received training, with 370 (96.9%) CHWs trained compared to 12 (3.1%) teachers trained (P<0.001). There was also a significant difference in education level, with 314 (91.5%) teachers having completed secondary education, compared to only 87(17%) among CHWs (P<0.001) (Table 1).

The mean age was 41.3, and the mean work experience was 10.1 years. (Table 1).

**Assessment of knowledge.** The overall mean knowledge score was 78%, with 461 (54.1%) of respondents who had a good knowledge level (score ≥80%). The mean knowledge score on STH (82.6%) was higher than on schistosomiasis (74.5%) (Table 2).

The three questions which the least respondents answered correctly were 1) "Pain during urination is a symptom of schistosomiasis" (32.2%), 2) "The most common symptom of schistosomiasis is fever" (58.6%) and 3) "Fever is not a common symptom of STH" (59.4%) (Table 2).

**Assessment of attitudes.** The three statements which most participants strongly agreed/agreed with were 1) "I am an important contributor to the prevention of schistosomiasis in my community" (97.2%); 2) "I am an important contributor to the prevention of STH in my community" (97.8%) and 3) "Deworming is helpful in treating schistosomiasis" (96.5%). The two statements which most participants strongly disagreed/disagreed with were 1) "STH can be best treated by traditional healers" (98.9%) and 2) "Schistosomiasis can be best treated by traditional healers" (98.5%) (Table 3).

**Assessment of practices.** Most respondents reported that they often or always wash hands before eating (n = 845, 99.2%) and often or always encourage children/people to wash their hands before eating (n = 844, 99.1%). The survey also showed 298 (35%) respondents reporting that they often or always drink untreated water, 790 (92.7%) never or rarely swim in rivers/lakes, and 771 (90.5%) never or rarely wash clothes or utensils in open water source (Table 4).

Most of respondents also reported drinking from treated water sources 822 (96.5%)—including tap water (52.1%) and borehole (44.4%). A small percentage of respondents drank from river or lakes (3.1%) and swamps or rocks (0.4%) (Table 5).

**Table 1. Social demographic characteristics of the study participants.**

| Characteristics | Variables | N (%) | CHWs | Teachers | P-value |
|---|---|---|---|---|---|
| Sample | | 852 | 509 (59.7%) | 343 (40.3%) | <0.001 |
| Age (year) | 18–35 | 257 (30.2%) | 84 (16.5%) | 173 (50%) | |
| | 36–49 | 419 (49.1%) | 279 (54.8%) | 140 (40.8%) | |
| | ≥50 | 176 (20.7%) | 146 (28.7%) | 30 (8.7%) | |
| | Mean (SD) | 41.3 (10.3) | 44.7 (9.1) | 36.2 (9.9) | |
| | Range | 20–70 | 24–70 | 20–65 | |
| Gender | Male | 310 (36.4%) | 173 (34%) | 137 (39.9%) | 0.082 |
| | Female | 542 (63.6%) | 336 (66%) | 206 (60%) | |
| District | Nyamagabe | 448 (52.6%) | 268 (52.7%) | 180 (52.5%) | 1.000 |
| | Rutsiro | 404 (47.4%) | 241 (47.3%) | 163 (47.5%) | |
| Marital status | Single | 77 (9.0%) | 4 (0.8%) | 73 (21.3%) | <0.001 |
| | Married | 746 (87.6%) | 479 (94%) | 267 (77.8%) | |
| | Divorced/widowed | 29 (3.4%) | 26 (5.1%) | 3 (0.9%) | |
| Highest Level of Education Attained | Primary | 400 (46.9%) | 399 (78%) | 1 (0.3%) | <0.001 |
| | Secondary | 401 (47.1%) | 87 (17%) | 314 (91.5%) | |
| | University | 28 (3.3%) | 1 (0.2%) | 27 (7.9%) | |
| | Vocational/ Literacy classes | 23 (2.7%) | 22 (4.3%) | 1 (0.3%) | |
| Years of experience | ≤ 5 | 292 (34.3%) | 171 (33.6%) | 121 (35.3%) | 0.612 |
| | ≥ 6 | 560 (65.7%) | 338 (66.4%) | 222 (64.7%) | |
| | Mean (SD) | 10.13 (7.6) | 9.61 (6.3) | 10.9 (9.2) | - |
| | Range | 0–44 | 0–35 | 0–44 | - |
| Training on worm infections | Trained | 382 (44.8%) | 370 (72.7%) | 12 (3.5%) | <0.001 |
| | Not trained | 470 (55.2%) | 139 (27.3%) | 331 (96.5%) | |
| Year when they received training on worm infections | 2016–2018 | 73 (19.1%) | 68 (18.4%) | 5 (41.7%) | 0.058 |
| | 2019–2021 | 309 (80.9%) | 302 (81.6%) | 7 (58.3%) | |

**Association of demographic characteristics and knowledge.** Five demographic factors (occupation, age, district, years of experience and training) in bivariate analysis had P<0.1. Only training in the deworming program was found to have a statistically significant association with knowledge using multivariate analysis. Participants who were trained were 2.04 more likely to have good knowledge on worm infections compared to those who were not trained (95% CI: 1.39–3.03, p <0.001) (Table 6).

**Association of knowledge and attitudes.** Knowledge level was found to have significant association with one attitude statement. Respondents with good knowledge level were found to be 4.66 times more likely to agree with the attitude statement "I am an important contributor to the prevention of schistosomiasis in my community" than those with poor knowledge, with a P<0.001 (Table 7).

**Association of knowledge and practices.** Knowledge level was found to be significantly associated with one practice statement. Respondents with good knowledge were 2.19 times more likely to "often/always" swim in rivers/lakes compared to those who had poor knowledge, with P = 0.006 (Table 8).

## Qualitative results

Seventeen leaders (10 from Nyamagabe and 7 from Rutsiro) were interviewed. Five key themes emerged:

**Table 2. Summary of knowledge about worm infections.**

| | Knowledge score | | |
| --- | --- | --- | --- |
| | **Total** | **CHWs** | **Teachers** |
| **Mean Schistosomiasis knowledge (SD)** | **74.5% (16.62)** | **74.9% (16.3)** | **73.8% (17.1)** |
| Schistosomiasis is treatable at the health center (Yes) | 87.3% | 85.3% | 90.4% |
| You can transmit schistosomiasis from open defecation (Yes) | 90.4% | 89.2% | 92.1% |
| Swimming in contaminated water can lead to schistosomiasis (Yes) | 95.1% | 96.3% | 93.3% |
| Washing clothes in contaminated water can lead to schistosomiasis (Yes) | 84.4% | 89.2% | 77.3% |
| You can get schistosomiasis from eating unripe fruits (No) | 71.1% | 69.2% | 74.1% |
| You can contract schistosomiasis through sexual intercourse (No) | 69.2% | 65% | 75.5% |
| Diarrhea is a symptom of schistosomiasis (Yes) | 82% | 77.6% | 88.6% |
| The most common symptom of schistosomiasis is fever (No) | 58.6% | 62.5% | 52.8% |
| Pain during urination is a symptom of schistosomiasis (Yes) | 32.2% | 40.3% | 20.1% |
| **Mean STH knowledge (SD)** | **82.6% (14.6)** | **83.4% (15)** | **81.6% (13.9)** |
| You can acquire STH from walking bare foot (Yes) | 94.4% | 94.9% | 93.6% |
| If you were cured from STH, you can never get it again (No) | 72.8% | 75.4% | 68.8% |
| Diarrhea is a symptom of STH (Yes) | 86.3% | 85.3% | 87.8% |
| Fever is the most common symptom of STH (No) | 59.4% | 63.2% | 55.1% |
| STH is hereditary (No) | 87.8% | 84.9% | 92.1% |
| You can acquire STH through direct skin contact (No) | 753 88.4% | 89% | 87.5% |
| You can get STH from drinking and eating contaminated water and food (Yes) | 89.4% | 91.7% | 86% |
| **Overall knowledge** | Mean score (SD) | 78% (12.44) | 78.6% (12.6) | 77.2% (12.1) |
| | Good knowledge (≥80%) | 461 (54.1%) | 288 (56.6%) | 173 (50.4%) |
| | Poor to moderate knowledge (<80%) | 391 (45.9%) | 221 (43.4%) | 170 (49.6%) |

The three questions that most respondents answered correctly were 1) "Swimming in contaminated water can lead to schistosomiasis" (95.1%), 2) "You can acquire STH from walking barefoot" (94.4%), and 3) "You can transmit schistosomiasis from open defecation" (90.4%).

## 1. Community mobilization and sensitization by local leaders and CHWs contributed to the outreach of the deworming program

In the national decentralized deworming program, local leaders are responsible for informing community members about the program and encouraging them to have their children

**Table 3. Summary of attitudes towards worm infections.**

| Attitude statements | Agreement (strong/regular) N (%) | Disagreement (strong/regular) N (%) |
| --- | --- | --- |
| Deworming is helpful in treating schistosomiasis | 822 (96.5%) | 30 (3.5%) |
| Deworming is helpful in treating STH | 791 (92.8%) | 61 (7.2%) |
| The people in my community are at high risk of acquiring schistosomiasis | 579 (68%) | 273 (32%) |
| The people in my community are at high risk of acquiring STH | 721 (84.6%) | 131 (15.4%) |
| I am an important contributor to the prevention of schistosomiasis in my community | 828 (97.2%) | 24 (2.8%) |
| I am an important contributor to the prevention of STH in my community | 833 (97.8%) | 19 (2.2%) |
| The frequency of mass drug administration against schistosomiasis in schools/community is enough | 247 (29%) | 605 (71%) |
| The frequency of mass drug administration against STH in schools/community is enough | 206 (24.2%) | 646 (75.8%) |
| Schistosomiasis can be best treated by traditional healers | 13 (1.5%) | 839 (98.5%) |
| STH can be best treated by traditional healers | 9 (1.1%) | 843 (98.9%) |

**Table 4. Summary of practices related to worm infections.**

| Practices | Often/always N (%) | Never/rarely N (%) |
|---|---|---|
| Encouraging children/ people to wash their hands | 844 (99.1%) | 9 (0.9%) |
| Washing hands before eating | 845 (99.2%) | 7 (0.8%) |
| Drinking untreated water | 298 (35%) | 554 (65%) |
| Swimming in rivers/ lakes | 62 (7.3%) | 790 (92.7%) |
| Washing clothes or utensils in open water source | 81 (9.5%) | 771 (90.5%) |

**Table 5. Sources of drinking water.**

| Water sources | | N (%) | |
|---|---|---|---|
| Good water source | Tap water | 444 (52.1%) | 822 (96.5%) |
| | Borehole | 378 (44.4%) | |
| Poor water source | Rivers or Lakes | 26 (3.1%) | 30 (3.5%) |
| | Swamps or Rocks | 4 (0.4%) | |

**Table 6. Association of demographic characteristics and knowledge.**

| | | Knowledge level | | Bivariate | Multivariate | |
|---|---|---|---|---|---|---|
| | | Good (≥80%) | Poor (<80%) | P-value | Adjusted OR (95%CI) | P-value |
| Occupation | Teachers | 173 (50.4%) | 170 (49.6%) | 0.078* | 0.79 (0.53–1.19) | 0.263 |
| | CHWs | 288 (56.6%) | 221 (43.4%) | | Ref | |
| Age | 18–35 | 133 (51.8%) | 124 (48.2%) | 0.094* | 0.96 (0.59–1.16) | 0.870 |
| | 36–49 | 220 (52.5%) | 199 (47.5%) | | 0.81 (0.56–1.18) | 0.264 |
| | ≥50 | 108 (61.4%) | 68 (38.6%) | | Ref | NA |
| Gender | Male | 166 (53.5%) | 144 (46.5%) | 0.804 | | |
| | Female | 295 (54.4%) | 247 (45.6%) | | | |
| District | Nyamagabe | 256 (57.1%) | 192 (42.9%) | 0.061* | 1.15 (0.87–1.53) | 0.334 |
| | Rutsiro | 205 (50.7%) | 199 (49.3%) | | Ref | NA |
| Marital status | Single | 42 (54.5%) | 35 (45.5%) | 0.879 | | |
| | Married | 402 (53.9%) | 344 (46.1%) | | | |
| | Divorced/ Widowed | 17 (58.6%) | 12 (41.4%) | | | |
| Education level | Primary | 228 (57%) | 172 (43%) | 0.403 | | |
| | Secondary | 205 (51.1%) | 196 (48.9%) | | | |
| | University | 16 (57.1%) | 12 (42.9%) | | | |
| | Vocational/Literacy classes only | 12 (52.2%) | 11 (47.8%) | | | |
| Years of experience | ≤ 5 | 145 (49.7%) | 147 (50.3%) | 0.060* | 0.806 (0.57–1.14) | 0.224 |
| | ≥ 6 | 315 (56.4%) | 244 (43.6%) | | Ref | NA |
| Training | Trained | 238 (62.3%) | 144 (37.7%) | <0.001* | 2.04 (1.39–3.03) | NA |
| | Not trained | 223 (47.4%) | 247 (52.6%) | | Ref | <0.001** |
| Years since worm program training | 2016–2018 | 46 (63%) | 27 (37%) | 0.889 | | |
| | 2019–2021 | 192 (62.1%) | 117 (37.9%) | | | |

*P<0.1

**P<0.05

**Table 7. Association of knowledge and attitudes.**

| | N Agree (%) | N Disagree (%) | Unadjusted OR | P-value |
|---|---|---|---|---|
| Deworming is helpful in treating schistosomiasis | | | | |
| Good knowledge | 450 (97.6) | 11 (2.4) | 2.09 (0.98–4.44) | 0.051 |
| Poor knowledge | 372 (95.1) | 19 (4.9) | Ref | |
| The people in my community are at high risk of acquiring schistosomiasis | | | | |
| Good knowledge | 319 (69.2) | 142 (30.8) | 1.13 (0.85–1.52) | 0.40 |
| Poor knowledge | 260 (66.5) | 131 (33.5) | ref | |
| I am an important contributor to the prevention of schistosomiasis in my community | | | | |
| Good knowledge | 456 (98.9) | 5 (1.1) | 4.66 (1.72–12.66) | <0.001* |
| Poor knowledge | 372 (95.1) | 19 (4.9) | ref | |
| The frequency of mass drug administration against schistosomiasis in schools/community is enough | | | | |
| Good knowledge | 134 (29.1) | 327 (70.9) | 1.01 (0.75–1.36) | 0.957 |
| Poor knowledge | 113 (28.9) | 278 (71.1) | ref | |
| Schistosomiasis can be best treated by traditional healers | | | | |
| Good knowledge | 9 (2%) | 452 (98) | 8.94 (0.59–6.29) | 0.402 |
| Poor knowledge | 4 (1%) | 387 (99) | ref | |
| Deworming is helpful in treating STH | | | | |
| Good knowledge | 426 (92.4) | 35 (7.6) | 0.87 (0.51–1.47) | 0.595 |
| Poor knowledge | 365 (93.4) | 26 (6.6) | ref | |
| The people in my community are at high risk of acquiring STH | | | | |
| Good knowledge | 392 (85%) | 69 (15%) | 1.07 (0.74–1.56) | 0.720 |
| Poor knowledge | 329 (84.1%) | 62 (15.9%) | ref | |
| I am an important contributor to the prevention of STH in my community | | | | |
| Good knowledge | 452 (98) | 9 (2%) | 1.32 (0.53–3.28) | 0.551 |
| Poor knowledge | 381 (97.4) | 10 (2.6) | ref | |
| The frequency of mass drug administration against STH in schools/community is enough | | | | |
| Good knowledge | 109 (23.6) | 352 (76.4) | 0.94 (0.69–1.29) | 0.693 |
| Poor knowledge | 97 (24.8) | 294 (75.2) | ref | |
| STH can be best treated by traditional healers | | | | |
| Good knowledge | 5 (1.1) | 456 (98.9) | 1.06 (0.28–3.98) | >0.999 |
| Poor knowledge | 4 [1] | 387 (99) | ref | |

dewormed. In addition, local leaders oversee and supervise the distribution of deworming tablets done by CHWs.

> "I work hand in hand with community health workers daily to make sure that every child of our Village receives the medicine that the government provided. I am present from the start till the end of the program. I have to make sure that all tablets are distributed, that they are over. . . yes, I have to be there." (IDI 3)

In case children are missing at the time of distribution, local leaders would conduct follow-up visits to the households to make sure that children receive their tablet.

> "At the sites, we checked out if parents brought children to get tablets as encouraged during household mobilization; and we could re-visit the ones we noticed did not come so that s/he brings the child the following day." (IDI 8)

**Table 8. Association of knowledge and practices.**

| | Often/ always n (%) | Never/rarely n (%) | Unadjusted OR | P-value |
|---|---|---|---|---|
| **Encouraging children/ people to wash their hands** | | | | |
| Good knowledge | 455 (98.7%) | 6 (1.3%) | 0.39 (0.78–1.93) | 0.300 |
| Poor knowledge | 389 (99.5%) | 2 (0.5%) | Ref | |
| **Washing hands before eating** | | | | |
| Good knowledge | 456 (98.9%) | 5 (1.1%) | 0.47 (0.00–2.40) | 0.462 |
| Poor knowledge | 389 (99.5%) | 2 (0.5%) | Ref | |
| **Drinking untreated water** | | | | |
| Good knowledge | 173 (37.5%) | 288 (62.5%) | 1.28 (0.92–1.70) | 0.090 |
| Poor knowledge | 125 (32%) | 266 (68%) | Ref | |
| **Swimming in rivers/ lakes** | | | | |
| Good knowledge | 44 (9.5%) | 417 (90.5%) | 2.19 (1.24–3.85) | *0.006 |
| Poor knowledge | 18 (4.6%) | 373 (95.4%) | Ref | |
| **Washing clothes or utensils in open water source** | | | | |
| Good knowledge | 44 (9.5%) | (90.5%) | 1.01 (0.64–1.60) | 0.968 |
| Poor knowledge | 37 (9.5%) | 354 (90.5%) | ref | |

## 2. Community members reportedly appreciate the value of the decentralized deworming program and express the desire to expand it

All 17 local leaders stressed that the community members appreciate the introduction of the deworming program at village level since it makes it easier for them to access the service without travelling long distances to health centers.

> *"They are very receptive because they do not walk miles to access the program. They are very receptive and happy with the program."* (IDI 1)

According to the local leaders interviewed, community members are asking for adult deworming tablets so that they too can benefit from the program. In addition, the local leaders express the need to increase the frequency of MDA in order to improve child health.

> *"My suggestion is. . . it is good that children get those deworming tablets. If advocacy can be done, adults also should be given tablets because they also have the same problem of worm infections. If there was the same program for adults, it could be good."* (IDI 5)

> *"My suggestion is that you can advocate so that the deworming tablets are available twice a year; that could be better. It should be twice a year and the distribution should continue to be done at the village level, because, when it was done at the cell level, it was not as effective as today at the village level. There was a crowd of people, and the distance was longer, and in some instances, all children could not be served in a single day and parents were discouraged as the process interfered with their daily activities. Today, they are happier to get the service at the village level as it is a decentralized program."* (IDI 1)

Local leaders also reported that they noticed good health outcomes in children as a result of the deworming program. According to them, it prevents children from falling frequently ill and being malnourished due to worm infestations.

> *"I mean, these tablets came on time because children had started to develop [enlarged] tummies, but after the deworming program, we can see the improvement. In fact, when the time of*

*the program approaches, people are ready to get their children to receive deworming tablets."* (IDI 16)

*"Yes, the impact is there because children have eliminated worms [in stool] and we realize a good growth development."* (IDI 16)

### 3. Complementary interventions to the deworming program were implemented by local leaders to prevent worm infections

All the local leaders reported that they worked in collaboration with CHWs to conduct community sensitization sessions on hygiene as well as household visits to monitor hygiene practices.

*"We encourage the community to always drink boiled water because unclean water is a host for worms. Another thing is. . .to encourage our people to have latrines, not to empty the bowels [defecate] in the outside in an open air because the waste contains worms that may get into the springs and rivers when it rains, and we may contract those when we use that water. Another thing is to encourage the community to have hygiene habits; hygiene of dishes, personal body hygiene and hygiene of foodstuff because poor hygiene in this stuff is also source of worm infections."* (IDI 17)

In view of improving household hygiene, community members, led by the local leaders, built adequate latrines for those who could not afford them.

*"For the households which have no latrines, we build those for them during umuganda [communal work in Rwanda] and the sector and district offices have a plan for that. The village community builds latrines, and the sector provides roofing materials." (IDI 17)*

### 4. Resistance and hesitance from caregivers are perceived by local leaders as challenges to the deworming program

Five local leaders observed that some caregivers do not adhere to the deworming program. Instead of bringing the children to the deworming sites, they would choose to go out to their farms.

*"When the community health workers come on time, they may also not find parents. . . that is the challenge. The parents come whenever they want, after farming activities."* (IDI 6)

According to one local leader, some caregivers are resistant to deworming because they think it increases food consumption by the children, which they cannot afford.

*"The main reason they say is that those deworming tablets have adverse effects to their children. Another reason they say is that when a child has taken that tablet, s/he eats a lot and "I will not be able to find that food."* (IDI 10)

According to some respondents, some caregivers perceive their children to be healthy, and they do not see the importance of deworming them.

*"There are no adverse effects, but the understanding of the community members is different. Some might say, "my child is healthy, s/he is not sick, so I am not taking him/her anyway."* (IDI 11)

### 5. Poor water access is identified by local leaders as a drawback in the prevention of worm infections

Twelve local leaders mentioned that their communities had poor access to safe water because of geographical and financial barriers, since they had to move long distances to access water sources and pay for the water.

"*About water. . .it is not yet fine, it is not yet good because, for instance in our village there is no drinking water, people still have to take jerry cans and go to fetch water; what we do is to encourage them to boil water so that it is safe to drink.*" *(IDI 5)*

"*Yes, it happens*! *It happens, the problem of low financial capacity exists, because when you tell a parent to give healthy food to their children or to adopt hygiene, it requires them to have money to buy soap or water, because in our village there is water available but to access it people have to pay for it. This problem of low financial means then arises.*" (IDI, 8)

Overall, the qualitative analysis led to the identification of key challenges based on the experiences and perspectives of local leaders, as well as their perspectives of the needs of the community.

## Discussion

CHWs, teachers and local leaders are key implementers of the national deworming program. The results of this study showed the knowledge of CHWs and teachers on worm infections could be improved. The community expressed the desire to increase the frequency of the deworming program and include adult members in the program.

Our results supported that training could improve the knowledge about deworming. Respondents who had received training had a higher knowledge on worm infections compared to those who had not. It is important to continue and scale up training efforts, especially when more than half of our participants (55.2%) had not received any prior training; particularly for teachers, as only 3.5% of them reported having received prior training. In 2018, about 42,000 CHWs were trained on common neglected tropical diseases, according to the Rwandan Biomedical Center [12]. Such training efforts should be expanded to teachers.

Regardless of training participation, the overall knowledge related to worm infections among teachers and CHWs was not good, with only 54% of the respondents having a good knowledge level. A previous study conducted in Ivory Coast had similar results, where respondents had low knowledge about worm infections [30]. Findings from our study showed that knowledge about symptoms and treatment of worm infections were low. Previous studies in Nigeria and Yemen also found that even if people had heard about schistosomiasis, their knowledge of symptoms was generally low [31,32]. Our findings support the need to strengthen deworming education for program implementers, with an emphasis on signs and symptoms of worm infections. The program curriculum should be reviewed and revised to ensure such knowledge gap is addressed. Improving this area of knowledge could potentially improve the ability of CHWs and teachers in early identification of infections in the community, which in turn could reduce the transmissibility within the community.

Our respondents generally held positive attitude towards the deworming program. CHWs and teachers were receptive to the program and believed it was helpful in preventing worm infections. The CHWs took on the responsibilities in the distribution of deworming tablets. They widely agreed they were important contributors in the prevention of worm infections, and the deworming was important. On this continent, the use of traditional medicine is widely

accepted in many countries [30]. However, most of our study respondents did not believe traditional healers could effectively treat worm infections. The receptive attitudes showed the success of the program in enhancing buy-in from stakeholders.

Overall, our study shows that the deworming program was appreciated by the local community leaders. Local leaders could identify the benefits of the decentralization of the MDA program, contributing to the ease of access to the tablets, and the numerous positive health outcomes. Similar appreciation of the decentralization of health services were seen in similar countries such as Kenya [33]. However, local leaders expressed a need to increase the frequency of MDA. Currently the WHO recommends frequency for MDA to be twice a year [34]. However, respondents perceived that some communities only had MDA once a year. Additionally, they expressed the need for adult deworming, as worm infections also affect adults. Studies have shown that adult who were engaged in farming activities could have higher risk than children of getting hookworm infections [35]. Policy makers should consider expanding the scope of the deworming program to include adults.

Local leaders reported that some misconceptions and resistance from the community members pose a challenge to the implementation of the deworming program. One of the misconceptions is that their children are healthy and do not need medication. Resistance to deworming is also found in other countries like Kenya, where misconceptions about the tablets was found to cause reluctance [33]. Additionally, according to local leaders, one of the reasons why community members exhibit resistance to the deworming program is their inability to provide enough food to their children, as they believe that deworming contributes to increasing children's appetite. This was highlighted in a similar study conducted in Kenya where food insecurity posed a major challenge to MDA compliance [33]. Misconceptions and resistance may become bottlenecks to the deworming program as they may hinder timely control and elimination of worm infections. Therefore, there is a need to strengthen community sensitization programs, with an emphasis on these misconceptions and on the impact of deworming on child health.

The study results showed that most of the respondents consumed water from taps and boreholes. However, due to geographical and financial limitations to water access, many people had to walk long distances to reach these water sources, and some were required to pay to access water in certain areas. Lack of access to safe water could increase the prevalence as well as the intensity of infection and re-infection [36]. Regardless, similar to many other neighboring countries, a considerable number of our respondents (298) representing 35% of respondents reported drinking untreated water [37]. Literature has suggested that effective deworming program must be complemented with water, sanitation and hygiene (WASH) interventions [38,39].

One unexpected finding in our study was that respondents with good knowledge on worm infections were more likely to swim in rivers/lakes. A study in Tanzania found people still had a tendency to visit the lakes/rivers for recreational purposes despite knowing the risk of infection from water bodies [40]. Studies have shown human decisions do not strictly follow norms, rules and regulations, but are influenced by a mixture of emotions, source of satisfaction and other psycho-social factors [41,42]. Further research to understand the decision process leading to the behavior is needed.

This study had some limitations which should be considered when interpreting its findings:

Firstly, the practice score in this survey relied on self-reporting. This could have subjected the findings to desirability bias, especially when practices were not directly observed. Secondly, this study only interviewed local leaders. Future study to collect community members' perspectives and experiences towards the deworming program or to indicate methods of improving/strengthening the program curriculum may generate further insights to complement our findings.

## Supporting information

**S1 Appendix. Quantitative data collection tool: This contains the questionnaire that was used for the quantitative survey.**
(DOCX)

**S2 Appendix. Qualitative data collection tool: This contains the questionnaire that was used for the qualitative interviews.**
(DOCX)

**S3 Appendix. Quantitative dataset: This contains the dataset that was generated from the quantitative survey and that was used to do the analysis.**
(XLSX)

## Acknowledgments

A special thanks goes to Dr. Rex Wong, Dr. Zahirah McNatt, and the entire faculty of the institute for Global Health Equity education at the University of Global Health Equity, for their support in the development of this research. We also thank Jessica Bourdaire, Grace Kelly Muvunyi, Georgette Munezero and Vera Kwera from the World Food Programme for their insightful feedback in the development of this manuscript.

## Author Contributions

**Conceptualization:** Fernand Rwamwejo, Grace Iliza Ndatinya, Madalitso Ireen Mkata, Amy Blauman, Denis Regnier, Sarah Cruz Lackner.

**Data curation:** Fernand Rwamwejo, Grace Iliza Ndatinya, Madalitso Ireen Mkata.

**Formal analysis:** Fernand Rwamwejo, Grace Iliza Ndatinya, Madalitso Ireen Mkata, Denis Regnier, Sarah Cruz Lackner.

**Investigation:** Fernand Rwamwejo, Grace Iliza Ndatinya, Madalitso Ireen Mkata.

**Methodology:** Fernand Rwamwejo, Grace Iliza Ndatinya, Madalitso Ireen Mkata, Sarah Cruz Lackner.

**Resources:** Sarah Cruz Lackner.

**Supervision:** Denis Regnier, Sarah Cruz Lackner.

**Writing – original draft:** Fernand Rwamwejo, Grace Iliza Ndatinya, Madalitso Ireen Mkata, Denis Regnier.

**Writing – review & editing:** Fernand Rwamwejo, Grace Iliza Ndatinya, Madalitso Ireen Mkata, Amy Blauman, Denis Regnier, Sarah Cruz Lackner.

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
