## [Decision Letter · Decision Letter 0]

8 Nov 2022

Dear Dr Rwamwejo,

Thank you very much for submitting your manuscript "Assessing the knowledge, attitudes, practices, and perspectives of stakeholders of the deworming program in rural Rwanda" for consideration at PLOS Neglected Tropical Diseases. As with all papers reviewed by the journal, your manuscript was reviewed by members of the editorial board and by several independent reviewers. The reviewers appreciated the attention to an important topic. Based on the reviews, we are likely to accept this manuscript for publication, providing that you modify the manuscript according to the review recommendations. 

The reviewers have taken time to suggest some relevant considerations for your paper. Overall, they have found the paper to be well-written and clearly outlined. Please be aware of their comments and address them in your response to reviewers as per PLoS NTD requirements. Some comments are directly embedded within the paper itself as comments, so please take note of those as well. 

A few additional points to consider here- 

Please note that there are papers that assessed the experiences of CHWs working in schistosomiasis campaigns (see the SCORE project papers) and examples from Kenya. While these may not be classified directly as KAP studies, they accomplish the same goals. (lines 21-23 of your paper). 

Consider revising terminology like 'impoverished populations' to 'people living in impoverished situations' which is more empowering language.

Sincerely,

Alison Krentel

Academic Editor

Francesca Tamarozzi

Section Editor

The reviewers have taken time to suggest some relevant considerations for your paper. Please be aware of their comments and address them in your response to reviewers as per PLoS NTD requirements. Some comments are directly embedded within the paper itself as comments, so please take note of those as well. 

A few additional points here- 

Please note that there are papers that assessed the experiences of CHWs working in schistosomiasis campaigns (see the SCORE project papers) and examples from Kenya. While these may not be classified directly as KAP studies, they accomplish the same goals. (lines 21-23 of your paper). 

Consider revising terminology like 'impoverished populations' to 'people living in impoverished situations' 

Overall, the paper is well-written and clearly outlined.

Reviewer's Responses to Questions

**Key Review Criteria Required for Acceptance?**

**Methods**

-Are the objectives of the study clearly articulated with a clear testable hypothesis stated?

-Is the study design appropriate to address the stated objectives?

-Is the population clearly described and appropriate for the hypothesis being tested?

-Is the sample size sufficient to ensure adequate power to address the hypothesis being tested?

-Were correct statistical analysis used to support conclusions?

-Are there concerns about ethical or regulatory requirements being met?

Reviewer #1: -Are the objectives of the study clearly articulated with a clear testable hypothesis stated? Yes

-Is the study design appropriate to address the stated objectives? Yes

-Is the population clearly described and appropriate for the hypothesis being tested? Yes

-Is the sample size sufficient to ensure adequate power to address the hypothesis being tested? Yes

-Were correct statistical analysis used to support conclusions? Yes

-Are there concerns about ethical or regulatory requirements being met? Not to me

Reviewer #2: The objectives are clearly stated.

The study design is quite appropriate to address the stated objectives

Sufficient sample size

Statistical analysis are okay to support conclusions

No such concerns

**Results**

-Does the analysis presented match the analysis plan?

-Are the results clearly and completely presented?

-Are the figures (Tables, Images) of sufficient quality for clarity?

Reviewer #1: -Does the analysis presented match the analysis plan? Yes 

-Are the results clearly and completely presented? Mostly. Please see recommendations in attached comments file.

-Are the figures (Tables, Images) of sufficient quality for clarity? Yes

Reviewer #2: The analysis presented correspond with the analysis plan. The results are clearly presented

The tables and figures are of sufficient quality for clarity. The variables in the tables should be refined to avoid repeatition

**Conclusions**

-Are the conclusions supported by the data presented?

-Are the limitations of analysis clearly described?

-Do the authors discuss how these data can be helpful to advance our understanding of the topic under study?

-Is public health relevance addressed?

Reviewer #1: -Are the conclusions supported by the data presented? Yes

-Are the limitations of analysis clearly described? Yes

-Do the authors discuss how these data can be helpful to advance our understanding of the topic under study? Yes they do, although please see comments for recommendation of additional further study that may be more directly contributive to the literature.

-Is public health relevance addressed? Yes, it is.

Reviewer #2: The conclusions are supported by the data presented. The limitations are clearly presented by the authors.

**Editorial and Data Presentation Modifications?**

Reviewer #1: A number of recommendations for clarity are addressed in the attached comments (copied below).

Introduction

• Line 71: is citation 13 for both statements?

• Line 75: missing a word before citation 15, “children infected with” what?

• Line 76: remove comma, also a very wide range?

• Line 79: “in conjunction” rather than “conjunct”

• Line 91: “and CHWs,”

Methods

• Line 109/110: possible duplication of “Sample”?

• Line 118: would it be possible to give some examples of what roles “local leaders” perform in the community? (i.e. civil, religious, etc.)

• Line 144: I believe this is the first mention of “cells”. There is a definition given later, all the way down in line 316 of the results. Please move this description to line 144 where “cell” is first used.

• Line 149: Please give a short description of what is meant by “theoretical saturation”, and a reference for “purposive sampling”.

• Line 158: “Informed consent was obtained”

• Line 174: a citation for the Likert scale may be appropriate https://psycnet.apa.org/record/1933-01885-001

• Line 186: What was the pre-testing mentioned here? Is this internal with the research team or were potential respondents shown the draft survey?

• Line 201: A reference for Fisher’s exact test may be appropriate https://link.springer.com/referenceworkentry/10.1007/978-3-642-04898-2_253

• Line 202/3: From this, it is hard to understand what attitude and practice categories really are. Describing the possible categories in the methods may be useful.

• Line 206: The statement says that all statistical tests were performed with a p-value of 0.05. This is confusing, is it perhaps the case that the authors mean that all results of the statistical tests with a p-value <0.05 were deemed to be significant?

• Good description of the translation/coding in lines 207-210, maybe a citation for Dedoose software would be appropriate.

• Line 212: Does the University of Global Health Equity IRB have an identification number or code?

Results

• Table 1: It may be more clear to indicate that “Education level” is the “Highest Level of Education Attained”. Otherwise some may be confused that only 1 Teacher had a primary level of education while 314 had a secondary level of education.

• Table 1: Is “Training in deworming” the same as “training on worm infections”? It is a little confusing to see them presented one above the other as though they are maybe the same.

• Table 1: I am a little confused about the p-values in table 1 that are seemingly comparing across three columns and 2 rows at the same time with the same value. For example, what is the p-value 0.058 in the “Year when they received training on worm infections” actually comparing? This may be my own misunderstanding, but it doesn’t seem that it should be exactly the same for both of the year categories.

• Table 3: The format makes it seem as though maybe only strongly agree and strongly disagree results are presented, but given that the total is 100% I assume the parentheses indicate these are combined across strongly/regular agreement. If they are to be combined, I would recommend changing the column headings to make this more clear, i.e. Agreement (strong/regular) and Disagreement (strong/regular), but I think it would be best to present both strong/regular and then combined as separate columns.

• Table 5: The second poor water source option is a little confusing to a reader not from this region. Is Swamps/Rock indicative of surface water from swampy/rocky areas? Or is “Rock” meant to indicate groundwater from a spring that comes out of rocks?

Discussion

• Line 394/395 is “knowledge about symptoms and treatment” from the Ivory coast study? If so, it would be clearer to combine this with the previous sentence/citation.

• It would be interesting and useful to see further research or an addition to this study that indicates some method of improving/strengthening the program curriculum (as recommended in line 399). Perhaps conjecture in this direction could be made in the limitations/further study section at the end of the discussion?

• Line 438 says that a significant portion (35%) reported drinking untreated water, with no p-value reported. Please include the significant p-value (i.e. (35%, p<0.xx)).

Reviewer #2: The manuscript should be accepted after minor revision.

**Summary and General Comments**

Reviewer #1: Overall, this study does a very fine job of highlighting the KAP of the study population with regards to deworming practices. Some of the presentation of the results could have been clearer (please see attached comments), but the overall message is quite clear, and the primary results stand out. I think the authors did an excellent job of including relevant qualitative data from the interviews, and in answering a relevant public health question. I think it would be very useful for the conclusions to suggest potential further studies that could actually tease out what sort of improved training materials actually may improve the knowledge scores of CHWs and teachers in this and similar geographies. No major revisions recommended.

Reviewer #2: The study was able to highlight the role of teachers and health workers in the deworming campaign, however there are critical stakeholders that were not factored in the study, which is the school age children, who are the direct beneficiaries of deworming interventions. Their perspectives and insights will provide added value to the study findings.

PLOS authors have the option to publish the peer review history of their article (what does this mean?). If published, this will include your full peer review and any attached files.

Reviewer #1: No

Reviewer #2: Yes: Obiageli Josephine Nebe

Figure Files:

Data Requirements:

Reproducibility:

References

---

## [Decision Letter · Decision Letter 1]

20 Mar 2023

Dear Dr Rwamwejo,

Thank you very much for submitting your manuscript "Assessing the knowledge, attitudes, practices, and perspectives of stakeholders of the deworming program in rural Rwanda" for consideration at PLOS Neglected Tropical Diseases. As with all papers reviewed by the journal, your manuscript was reviewed by members of the editorial board and by several independent reviewers. The reviewers appreciated the attention to an important topic. Based on the reviews, we are likely to accept this manuscript for publication, providing that you modify the manuscript according to the review recommendations. 

Sincerely,

Alison Krentel

Academic Editor

Francesca Tamarozzi

Section Editor

Reviewer's Responses to Questions

**Key Review Criteria Required for Acceptance?**

**Methods**

-Are the objectives of the study clearly articulated with a clear testable hypothesis stated?

-Is the study design appropriate to address the stated objectives?

-Is the population clearly described and appropriate for the hypothesis being tested?

-Is the sample size sufficient to ensure adequate power to address the hypothesis being tested?

-Were correct statistical analysis used to support conclusions?

-Are there concerns about ethical or regulatory requirements being met?

Reviewer #1: Yes, methods strong throughout the process.

Reviewer #2: The objectives of the study were clearly stated. The study design was found to be appropriate. The population was clearly stated but the study would have included school age children who are one of the critical stakeholders and beneficiaries of deworming Programme. Their insights and perspectives on deworming activities are equally needed and will make important contributions in the study of this kind. I think the sample size is adequate and equally addressed the hypothesis being tested.

Correct statistical analysis and variables were used to support the study.

There are no known ethical concerns as the study did not involve any invasive procedures.

**Results**

-Does the analysis presented match the analysis plan?

-Are the results clearly and completely presented?

-Are the figures (Tables, Images) of sufficient quality for clarity?

Reviewer #1: Yes, changes were made to have the tables be a little clearer to read and understand.

Reviewer #2: The analysis presented matched the analysis plan. The results are clearly presented.

The figures and tables are of sufficient quality.

**Conclusions**

-Are the conclusions supported by the data presented?

-Are the limitations of analysis clearly described?

-Do the authors discuss how these data can be helpful to advance our understanding of the topic under study?

-Is public health relevance addressed?

Reviewer #1: Yes, the conclusion is well written and clearly addresses limitations and use of these data for advancing the knowledge base.

Reviewer #2: The authors have to clearly describe and highlight the conclusions emanating from this study. They need to further describe the limitations of the analysis if any. I did not see in the manuscript, major recommendations and implications of the findings of the study to the National Deworming Programnme of Rwanda. By clearly describing how the outcomes of the study will impact on the Deworming Programme and may contribute to informed-decision making/policy change, this will bring to fore the public health relevance of the study.

**Editorial and Data Presentation Modifications?**

Reviewer #1: All concerns and comments from the first submission have been satisfactorily addressed.

Reviewer #2: I recommend that the minor revision be done, and that the authors should carefully address the comments made in the draft manuscript.

**Summary and General Comments**

Reviewer #1: (No Response)

Reviewer #2: My overall comment is that the authors should address the gaps highlighted in the manuscript. One of the key beneficiaries of deworming Programme which are school age children were left out of this study. The authors should find away to interview them and incorporate their views and perspectives into the finding.

PLOS authors have the option to publish the peer review history of their article (what does this mean?). If published, this will include your full peer review and any attached files.

Reviewer #1: No

Reviewer #2: Yes: Obiageli Josephine Nebe

Figure Files:

Data Requirements:

Reproducibility:

References

---

## [Decision Letter · Decision Letter 2]

17 Jul 2023

Dear Dr Rwamwejo,

We are pleased to inform you that your manuscript 'Assessing the knowledge, attitudes, practices, and perspectives of stakeholders of the deworming program in rural Rwanda' has been provisionally accepted for publication in PLOS Neglected Tropical Diseases.

Best regards,

Francesca Tamarozzi

Section Editor

Francesca Tamarozzi

Section Editor

Reviewer's Responses to Questions

**Key Review Criteria Required for Acceptance?**

**Methods**

-Are the objectives of the study clearly articulated with a clear testable hypothesis stated?

-Is the study design appropriate to address the stated objectives?

-Is the population clearly described and appropriate for the hypothesis being tested?

-Is the sample size sufficient to ensure adequate power to address the hypothesis being tested?

-Were correct statistical analysis used to support conclusions?

-Are there concerns about ethical or regulatory requirements being met?

Reviewer #2: The objectives of the study was clearly articulated with a clear testable hypothesis stated by the authors. The study design was quite appropriate and addressed the stated objectives. The use of phenomenological design to explore the stakeholders experiences and perspectives was aptly stated. The study population was clearly described and quite appropriate for the hypothesis tested. There is sufficient sample size, and ensure adequate power for the hypothesis tested. Correct statistical analysis were used to support the conclusion. The use of Karimollah formular for sample size estimation was good great and will be of interest to the scientific community. No ethical concerns whatso ever.

**Results**

-Does the analysis presented match the analysis plan?

-Are the results clearly and completely presented?

-Are the figures (Tables, Images) of sufficient quality for clarity?

Reviewer #2: The analysis presented really match the analysis plan. The use of Karimollah formular for sample size estimation was good great and will be of interest to the scientific community. The results are clearly and completely presented. The tables are of sufficient quality for clarity.

**Conclusions**

-Are the conclusions supported by the data presented?

-Are the limitations of analysis clearly described?

-Do the authors discuss how these data can be helpful to advance our understanding of the topic under study?

-Is public health relevance addressed?

Reviewer #2: The conclusions were supported by the data presented. The limitations were equally stated. The authors discussed how the data will help in advancing the understanding of the topic that was studied. Public health relevance was addressed.

**Editorial and Data Presentation Modifications?**

Reviewer #2: I hereby recommend that the manuscript be accepted after minor revision as stated in the manuscript.

**Summary and General Comments**

Reviewer #2: The manuscript is well articulated, the study design- the use of cross-sectional quantitative and phenomenological design to explore the experiences and perspectives of stakeholders is quite significance. The data plan and analysis especially the use of Karimollah formular and details of the statistical formular for sample size estimation was great. No new experiment required. The research was conducted in accordance with the principles expressed in the Declaration of Helsinki. The manuscript was written in intelligible fashion and standard English. Accept the manuscript after minor revision.

PLOS authors have the option to publish the peer review history of their article (what does this mean?). If published, this will include your full peer review and any attached files.

Reviewer #2: **Yes: **Dr. Obiageli Josephine Nebe

---

## [Editor Report · Acceptance letter]

3 Aug 2023

Dear Dr Rwamwejo,

We are delighted to inform you that your manuscript, "Assessing the knowledge, attitudes, practices, and perspectives of stakeholders of the deworming program in rural Rwanda," has been formally accepted for publication in PLOS Neglected Tropical Diseases.

Best regards,

Shaden Kamhawi

co-Editor-in-Chief

Paul Brindley

co-Editor-in-Chief
